# Self-Control, Openness, Personal Need for Structure and Compensatory Control Change: A Serial Mediation Investigation

**DOI:** 10.3390/bs14050352

**Published:** 2024-04-23

**Authors:** Yuan Zhao, Yuying Wang, Liuqing Wei, Yu Ma, Yunyun Chen, Xuemin Zhang

**Affiliations:** 1Beijing Key Laboratory of Applied Experimental Psychology, Faculty of Psychology, National Demonstration Center for Experimental Psychology Education, Beijing Normal University, Beijing 100875, China; 202121061011@mail.bnu.edu.cn (Y.Z.); yuying.wang@student.kuleuven.be (Y.W.); 202031061006@mail.bnu.edu.cn (Y.C.); 2Faculty of Psychology and Educational Sciences, KU Leuven, 3000 Leuven, Belgium; 3Department of Psychology, Faculty of Education, Hubei University, Wuhan 430062, China; 20180084@hubu.edu.cn; 4Center for Psychological Health, Ningxia Vocational Technical College of Industry and Commerce, Ningxia 750021, China; 201831061066@mail.bnu.edu.cn; 5State Key Laboratory of Cognitive Neuroscience and Learning, Beijing Normal University, Beijing 100875, China

**Keywords:** COVID-19, self-control, compensatory control, openness, personal need for structure

## Abstract

In the aftermath of the COVID-19 pandemic, numerous studies have indicated that individuals are confronting a diminished sense of control. Compensatory control theory suggests that individuals strive to mitigate this loss by modifying their behavior. The present study aims to investigate the relationship between *self-control* and *compensatory control change* during the COVID-19 pandemic, as well as the mediating effects of openness and the personal need for structure. Participants completed an online questionnaire consisting of Personal Need for Structure Scale, Self-Control Scale, Openness Scale and Compensatory Control Change Scale. The results showed that the *compensatory control change* increased after the outbreak. Moreover, a serial mediation was found: *openness* and the *personal need for structure* partially mediated the relationship between *self-control* and *compensatory control change*. The results indicate that the COVID-19 pandemic has led to an increase in compensatory control behaviors, especially among those with pronounced self-control. High self-control individuals are found to exhibit greater openness, reducing their personal need for structure, in effect enhancing their compensatory control change. These findings highlight the critical role of self-control in sustaining a sense of control, which is vital for understanding psychological health management in the context of public health events.

## 1. Introduction

During the COVID-19 pandemic, various aspects of people’s lives underwent significant changes, such as the shift to working from home and reduced travel, which subsequently led to an increase in mental health issues [1,2]. While the COVID-19 pandemic has become a thing of the past, its potential enduring negative impacts on human behavior and psychological well-being cannot be overlooked. For example, research across different countries indicated that the outbreak has resulted in common psychological issues, such as depression and anxiety, among the population [3,4,5,6]. The pandemic’s widespread effects extended beyond immediate health concerns, disrupting daily routines, social interactions and economic stability, further exacerbating feelings of isolation and uncertainty. These factors collectively contributed to the observed surge in mental health challenges, underscoring the complex interplay between external disruptions and internal psychological processes.

When addressing mental issues caused by public health events, such as COVID-19, it is crucial to gain in-depth insight into some key reasons behind their emergence. Notably, the profound environmental changes induced by the outbreak left individuals feeling powerless, triggering a lack of sense of control [7,8,9,10]. Sense of control, the belief that one’s actions or personal characteristics can influence one’s environment or outcomes [11], is closely linked to psychological health. Boosting one’s sense of control can enhance psychological resilience [12] and potentially reduce psychological stress [13,14]. Understanding and addressing the underlying factors that diminish the sense of control can, therefore, be a vital strategy in mitigating the long-term psychological impacts of the pandemic, promoting recovery and preparing for future public health challenges.

### 1.1. The Compensatory Control Theory

The absence of a sense of control can generate compensatory control behaviors. Kay et al. [15] introduced the Compensatory Control Theory (CCT), which posits that people’s confidence in life depends on their perception of the external world as structured and controllable, rather than chaotic. Consequently, individuals experiencing prolonged periods of low control tend to seek structure and consistency [10,15,16]. This inclination is reflected in a range of compensatory behaviors that aim to introduce order and structure into the environment, such as embracing conspiracy beliefs, preference for order-providing scientific theories, moral arguments and irrational hoarding [17,18,19]. During the pandemic, several compensatory control behaviors were observed, including the sharing of conspiracy theories on social media, believing and spreading misinformation about the virus and stockpiling food and toiletries [20,21,22].

### 1.2. The Trait Self-Control and Associated Factors

When facing significant life events like the COVID-19 outbreak, people’s ability to compensate for their sense of control through behaviors aimed at controlling their environment is limited. Additionally, individuals’ inherent trait of self-control plays a crucial role in their sense of control [23]. Self-control, as a trait, often represents a crucial resource for affecting personal life outcomes and is a key factor in addressing personal and societal issues [24,25]. Evidence from various studies supports this, showing that self-control acts as a reliable indicator of an individual’s ability to curb impulsive or irrational behaviors, as well as achieve academic success [26,27,28].

The level of the trait self-control significantly influences how individuals respond to a change in environment. Regarding the essential factor of environmental order, research has shown that participants with lower self-control demonstrated enhanced self-regulatory ability in structured environments than in chaotic ones, whereas those with higher self-control were less affected by environmental conditions [29]. Moreover, several studies indicated that lower self-control correlated with a higher personal need for structure (PNS) [30] and stronger compensatory control [16,18]. This difference may be linked to the way individuals with higher self-control predominantly derive their control internally, resulting in a consistently lower PNS and less engagement in compensatory behaviors, regardless of the environmental context. Conversely, those with lower self-control tend to seek more structure and are more prone to compensatory behaviors, particularly in less structured environments. Moreover, given these observations, a PNS might act as a mediator between the trait self-control and compensatory control.

Individual reactions to significant life events are based on their personality traits. Importantly, traits such as openness significantly determine how individuals cope with uncertainty and ambiguity. Research by Neuberg and Newsom [31] suggested a negative correlation between PNS and openness. Similarly, Ma and Kay’s [10] study on ambiguity and uncertainty was consistent with this finding. It implied that individuals with higher PNS tend to have lower tolerance for ambiguity and uncertainty, whereas those with greater openness demonstrated a heightened tolerance for chaos and ambiguity [32]. Hence, when exploring the correlation between the trait self-control and compensatory control, openness may emerge as a crucial factor that cannot be overlooked. It may interact with the PNS to jointly influence the correlation between the trait self-control and compensatory control.

### 1.3. The Hypotheses of the Present Study

Building on these above insights, the present study aims to explore the relationship between self-control and compensatory control change (specifically, the differences in compensatory control before and after the outbreak) triggered during the COVID-19 pandemic, as well as the mediating roles of openness and PNS. As previously mentioned, our focus is primarily on several key points. Firstly, the COVID-19 pandemic has intensified social disorder. Secondly, compared to individuals with higher self-control, those with lower levels may exhibit greater variability in compensatory control behaviors across different environments. Thirdly, openness and the PNS may jointly mediate the correlation between the trait self-control and compensatory control.

We hypothesized that (I) the COVID-19 pandemic has led to an increase in compensatory control; (II) the trait self-control (in the later text, the trait self-control will be referred to as self-control) is negatively associated with both compensatory control change (in the later text, compensatory control change will be referred to as CCC) and a PNS but positively associated with openness; (III) openness is negatively associated with CCC, but the PNS is positively associated with CCC; and (IV) openness and a PNS would mediate the relationship between self-control and CCC. The participants of this study are college students, as some research indicates that young adults aged 18–34 are at high risk for pandemic-related psychological health deterioration and exhibit more compensatory control behaviors [21,33,34].

## 2. Materials and Methods

### 2.1. Participants

The sample size was determined via a prior power analysis using G*Power software, version 3.1.9.7 [35]. The power analysis indicated that the minimum sample size was 129 for a statistical power of 0.95, assuming a Type I error probability of 0.05 and medium effect size (f = 0.15). To ensure sample diversity, we collected data from 181 college students across three Chinese universities (100 females, 81 males; *M*_age_ = 23.5, *SD*_age_ = 2.02, range from 18 to 36) who completed an online survey via a questionnaire platform https://www.wjx.cn/ (accessed on 11 July 2022).

All participants were enrolled in school by 2019 and had at least one year of college campus experience before the outbreak. They gave their consent to participate in this study and were informed that their responses would be anonymous. Moreover, each participant received CNY 3 for the time.

### 2.2. Measures

#### 2.2.1. Personal Need for Structure Scale (PNS)

The PNS measures the psychological need to perceive one’s existence and surroundings as clear, orderly, predictable and not ambiguous or random [36]. Neuberg and Newsom [31] revised the PNS scale, which consisted of 11 items, with a median Cronbach’s α rating of 0.77. Chen et al. [37] carried out the Chinese version of the PNS with well-calibrated correlation validity, internal consistency and split-half reliability (Cronbach’s α = 0.82). The PNS is a 6-point Likert scale with total scores ranging from 11 to 66. Higher scores indicate higher structural demands. For more details, please see Appendix A.

#### 2.2.2. Self-Control Scale (SCS)

The SCS is a self-report measure used to assess individual differences in self-control traits. It measures self-control as a dispositional characteristic and provides information about a person’s ability to exercise self-control [38]. Unger et al. [39] revised the Chinese version of the SCS scale, including the Full Chinese Self-Control Scale (36 items, Cronbach’s α = 0.88) and the Brief Chinese Self-Control Scale (12 items, Cronbach’s α = 0.75). We used the Brief version, which is a 5-point Likert scale. The total score ranges from 12 to 60, with higher scores indicating greater self-control. For more details, please see Appendix A.

#### 2.2.3. Openness Scale (OS)

Openness reflects curiosity, creativity, resourcefulness and a willingness to consider unconventional ideas [40]. We used the openness subscale from the Chinese Big Five Personality Inventory Brief Version (CBF-PI-B, Cronbach’s α = 0.79) [41] to assess Chinese people’s openness. The test-retest reliabilities 10 weeks after of openness was 0.81. There are 8 items on a 6-point Likert scale (1 = strongly disagree, 6 = strongly agree). The total score ranges from 8 to 48, with higher scores indicating greater openness. For more details, please see Appendix A.

#### 2.2.4. Compensatory Control Change Scale (CCC)

The CCC includes five types of questions related to compensatory control, covering hoarding, consumption, travel, goals and job expectations. Each type contains two questions concerning compensatory control before and after the outbreak. The CCC aims to examine whether there was a significant increase in compensatory control after the outbreak. Questions are presented as follows (All of option B represent compensatory control and vice versa for option A):

1. Did/do you have a habit of hoarding before/after the outbreak? (A. no; B. yes)

2. Your primary reason for shopping before/after the outbreak: (A. I want; B. I Need)

3. Your trip packing habits before/after the outbreak: (A. I prefer to go light and pack as little as possible; B. I prefer to be well prepared and bring as much as possible)

4. Did/do you set long-term goals before/after the outbreak? (A. I am more future-oriented, setting long-term objectives and working hard to achieve them; B. I am more present-oriented and attempt to do the right thing now)

5. Before/after the outbreak, what was your ideal occupation was? (A. IT Industry/self-employment/other; B. Further education/civil service/state-owned enterprises or institutions)

In each question, option A is assigned a value of 1, while Option B is assigned a value of 2. The score of the CCC is the difference between before and after the outbreak (the value after the outbreak minus the value before the outbreak), with a total score of five questions ranging from −5 to 5. The positive score indicates more compensatory control behaviors after the outbreak than before, whereas the negative value indicates the opposite.

### 2.3. Data Analyses

Data analysis was performed using SPSS Statistics 26.0. We examined the descriptive statistics of each variable and the correlations between them. The data normality was assessed before mediation analyses. The skewness of the variables varied between −0.30 and 0.34, while their kurtosis varied between −0.75 and −0.02, demonstrating that the variables met the normality criterion. The Durbin–Watson value was ±1.96, with variance inflation factor values being between 1.04 and 1.14 and the tolerance value being between 0.88 and 0.96. When the results above considered the recommendations of Chapman [42], it was established that multicollinearity and residual problems were not present. The mediation analyses were conducted using the SPSS PROCESS macro [43] to test the hypothesized model that openness and PNS mediate the relationship between self-control and CCC, with age and gender as covariate variables. A bootstrap sample of 5000 was utilized in the analysis, with a 95% confidence range. If zero was not included in the lower and upper bound interval, the effect was statistically significant at *p* < 0.05.

## 3. Results

### 3.1. Preliminary Analyses

Table 1 displays the means and standard deviations of the questions of compensatory control. The mean score of the CCC was positive (*M* = 1.27, *SD* = 1.26). A paired-sample *t*-test of the total score of five types of questions revealed that compensatory control was significantly different. It was greater after the outbreak than before the outbreak (*t* = −13.62, *p* < 0.001, *d* = 1.01). Specifically, participants reported more compensatory control after versus before the outbreak for almost every type of question (paired-sample *t*-test of the five types of questions: hoarding; *t* = −7.84, *p* < 0.001, *d* = 0.58; consumption: *t* = −9.13, *p* < 0.001, *d* = 0.67; travel: *t* = −7.16, *p* < 0.001, *d* = 0.54; goals: *t* = −5.07, *p* < 0.001, *d* = 0.37; job expectations: *t* = −0.001, *p* < 0.05, *d* = 0.16). These results demonstrated that there were more behaviors of compensatory control after the outbreak.

The correlation analysis exhibited that self-control, CCC, openness and a PNS were significantly inter-related. Self-control, openness and CCC were negatively related to PNS, while self-control, openness and CCC were positively correlated with each other (see Table 2). Additionally, a significant correlation between gender and CCC was found, with females exhibiting more remarkable CCC than males (*r* = 0.18, *p* < 0.05).

### 3.2. Mediation Analyses

Figure 1 exhibits the results of mediation analyses. Self-control demonstrated a direct positive effect on CCC (total effect: *β* = 0.226, 95% CI = [0.089, 0.363]). When openness and PNS were included as mediators, the direct effect of self-control on CCC was slightly reduced (direct effect: *β* = 0.176, 95% CI = [0.041, 0.311]), indicating partial mediation. Specifically, self-control was a direct positive predictor of openness (*β* = 0.154, 95% CI = [0.009, 0.299]), which was a direct negative predictor of PNS (*β* = −0.296, 95% CI = [−0.433, −0.158]) and direct positive predictor of CCC (*β* = 0.147, 95% CI = [0.005, 0.289]). Each path within the mediation reflects a specific hypothesized psychological process that contributes to the overall effect on CCC. While self-control did not significantly predict PNS directly (*β* = 0.113, 95% CI = [−0.249, 0.024]), PNS emerged as a significant negative predictor of CCC (*β* = −0.175, 95% CI = [−0.320, −0.029]). This result suggests that self-control impacts CCC primarily through openness (see Table 3). Further analysis revealed that the combined indirect effect of self-control on CCC through the sequential mediation of openness, followed by PNS, was statistically significant (*β* = 0.008, 95% CI = (0.000, 0.023]) (see Table 3).

## 4. Discussion

The current study aimed to explore the influence of social disruption resulting from public health emergencies on the individual’s sense of control and corresponding behaviors. Specifically, it investigated whether college students exhibited increased compensatory control behaviors in the context of COVID-19 and demonstrated that openness and the personal need for structure (PNS) serve as mediators between self-control and the compensatory control change (CCC).

### 4.1. The Impact of the Outbreak on Compensatory Control Behaviors

Compensatory control serves as a crucial psychological strategy, enabling individuals to regain a sense of agency and stability when facing situations that threaten their perceived control over their environment [15,16,44]. Our study specifically examined how the onset of the COVID-19 pandemic impacted compensatory control behaviors among college students, revealing a significant uptick in these behaviors following the outbreak. This finding suggests that the pandemic’s disruption to daily life and the ensuing uncertainty significantly challenged students’ sense of control, prompting them to engage more extensively in behaviors aimed at restoring it. These behaviors are manifested in multiple aspects closely related to them, including hoarding, consumption, travel, goals and job expectations.

As a matter of fact, the adoption of compensatory control behaviors in response to the pandemic is reflective of a broader psychological mechanism where individuals strive to counterbalance perceived threats to their control. This mechanism underscores the adaptive nature of human behavior in the face of crises, as individuals seek to maintain or restore psychological equilibrium.

### 4.2. The Serial Mediation of Self-Control, Openness, PNS and CCC

The current study has found that openness and the PNS serve as partial mediators in the relationship between self-control and CCC. Individuals with high self-control tend to exhibit greater openness, leading to a reduced PNS yet an increased CCC in the context of the pandemic.

The correlational analysis revealed a positive association between self-control and openness, with both variables demonstrating a negative correlation with the PNS. Consequently, individuals exhibiting higher levels of self-control tend to possess greater openness and exhibit lower PNS, aligning with our initial hypothesis. However, contrary to our prediction, self-control and openness were found to positively correlate with CCC, whereas PNS negatively correlated with CCC. Specifically, individuals with higher self-control exhibited a more substantial increase in compensatory control behaviors after the outbreak, in contrast to those with lower self-control. Likewise, individuals with greater openness experienced an increase in such behaviors more so than the less open ones. However, those with higher PNS experienced a smaller uptick compared to those with lower PNS. Indeed, the discrepancies observed primarily arise from the correlation between self-control and CCC, while the correlations among self-control, openness and personal need for structure (PNS) remain consistent with anticipated outcomes.

Our initial prediction was that individuals with lower self-control would have higher CCC. The unexpected results may suggest that individuals with higher self-control maintain better stability in their sense of control through enhanced self-regulation, even in less structured or chaotic environments. This interpretation aligns with the findings of Lindner et al. [45], who noted that individuals with high self-control were associated with lower levels of perceived risk of infection, which was further linked to reduced health anxiety. Such results underline the role of self-control in minimizing exposure to risky situations and managing health-related fears during the COVID-19 pandemic. Individuals with high self-control not only thrive during significant life events but also exhibit more adaptive behavior in normal situations. Cheung et al. [46] observed that these individuals are more promotion-focused, aiming to acquire positive gains that facilitate approach-oriented behaviors. Conversely, they are less prevention-focused, showing fewer avoidance-oriented behaviors. Pfeffer and Strobach [47] found that a higher level of self-control is associated with a reduced gap between intentions and behaviors. This finding implies that individuals possessing greater self-control are more effective at implementing their intended plans and achieving their set goals. It is noteworthy that the findings of this study, demonstrating a positive correlation between openness and CCC and a negative correlation between PNS and CCC, further substantiate that individuals with high self-control exhibit enhanced adaptability to environmental changes. Therefore, individuals with high self-control not only tend to possess greater openness but are also more capable of tolerating the uncertainties associated with chaotic environments.

In a word, these observations compellingly demonstrate that individuals with high self-control possess robust self-regulation capabilities, enabling them to effectively manage various life events, especially significant ones. Additionally, several studies have shown that high self-control also enhances emotional regulation. Individuals with high self-control not only tend to engage in more spontaneous emotion regulation but also report higher levels of well-being [46,48,49]. Further scrutiny of how self-control affects emotion regulation is necessary. It was found that individuals with lower self-control exacerbated their responses to uncertainty and potentially raised their stress levels due to reduced compliance with health guidelines during the COVID-19 pandemic [50]. Furthermore, those with diminished self-control often exhibit greater psychological vulnerability and require more support to maintain good mental health [51]. Therefore, understanding the dynamics of self-control is crucial for developing effective mental health interventions in response to public health emergencies.

### 4.3. Limitations, Implications and Future Directions

The present study, while offering valuable insights, encounters certain limitations that merit attention. Firstly, the participants were limited to Chinese college students, potentially narrowing the ecological validity and applicability of the findings across different cultural and age demographics. Secondly, the data were gathered at one time point, relying on retrospective assessment of “before the outbreak” behaviors, which introduces the possibility of recall bias due to inaccuracies in participants’ memories of their pre-pandemic actions. Furthermore, the use of cross-sectional data to conduct mediation analysis limits the ability to make causal claims, as Kline [52] notes, due to the inability to establish the necessary temporal sequence among variables for confirming mediation. These issues highlight the need for careful interpretation of the mediation effects observed between self-control, openness, PNS and CCC.

The implications of our findings underscore the critical importance of fostering a sense of control through individual self-regulation and environmental mastery. In the face of public health crises or major societal disruptions, governments and community organizations must recognize the significant role that a stable and orderly social and digital environment plays in reducing psychological distress. This recognition becomes even more critical for individuals with lower levels of self-control. To mitigate adverse effects, it is imperative to adopt strategies that halt the dissemination of misinformation related to COVID-19 and ensure resource accessibility, thereby minimizing the reduction in people’s sense of control caused by environmental changes. Furthermore, establishing accessible psychological support online and offline is essential for providing immediate and effective aid to those in need.

Future studies should give priority to longitudinal designs that broaden the diversity of participants, aiming to shed light on the causal links between self-control, openness, PNS and compensatory control behaviors. In scenarios like public health emergencies or significant life events, researchers are encouraged to conduct follow-up studies that explore how individuals resort to compensatory control strategies, considering variables such as trait self-control, openness and PNS. Cross-lagged regression analysis could then be utilized to define the causal relationships between these factors. Furthermore, delving into potential mediators or moderators and the effects of previously unexamined confounding variables would deepen our understanding of the intricate dynamics involved. Tackling these areas is essential for crafting well-rounded psychological support strategies to effectively meet the challenges of public health events.

## 5. Conclusions

The present study offers valuable insights into the dynamics of sense of control. It was observed that post-outbreak, college students exhibited increased compensatory control behaviors compared to the pre-outbreak period. Furthermore, it was found that openness and PNS serve as partial mediators in the interaction between self-control and CCC. Specifically, individuals with higher self-control tended to exhibit greater openness, which, in turn, reduced their need for structure and led to an increase in CCC. These outcomes underscore the crucial role of high self-control in maintaining and enhancing a sense of control. These findings provide a foundation for developing targeted interventions that leverage self-control to bolster resilience and adaptive responses in the face of public health events. By understanding the mediating effects of openness and PNS, future interventions can more effectively support individuals when navigating the psychological challenges posed by such events, ultimately contributing to improved mental health and well-being.

## Figures and Tables

**Figure 1 behavsci-14-00352-f001:**
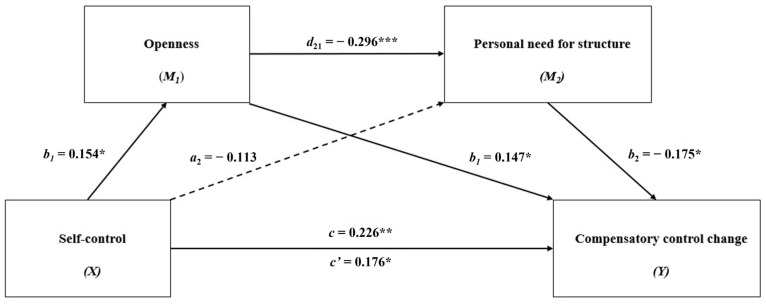
The result of the serial multiple mediational model: * *p* < 0.05, ** *p* < 0.01, *** *p* < 0.001. The values shown are unstandardized coefficients.

**Table 1 behavsci-14-00352-t001:** Descriptive statistics of compensatory control.

	Before the Outbreak		After the Outbreak	
	*M*	*SD*	*M*	*SD*
Hoarding	1.38	0.49	1.67	0.47
Consumption	1.27	0.44	1.63	0.48
Travel	1.23	0.42	1.55	0.50
Goals	1.36	0.48	1.59	0.49
Job expectation	1.77	0.42	1.84	0.37
Total score	7.01	0.96	8.28	1.16

**Table 2 behavsci-14-00352-t002:** Descriptive statistics and correlations among variables.

Variables	1	2	3	4	5	6
1. Personal need for structure (PNS)	——					
2. Self-control	−0.162 *	——				
3. Openness	−0.333 ***	0.155 *	——			
4. Compensatory control change (CCC)	−0.265 ***	0.239 **	0.253 **	——		
5. Gender	−0.020	0.041	0.053	0.177 *	——	
6. Age	0.133	0.005	−0.126	−0.097	−0.028	——
*M* *SD*	47.88 6.49	39.06 8.52	33.98 6.77	1.27 1.26	1.55 0.50	23.53 2.02
Skewness	−0.295	0.236	−0.203	0.343		
Kurtosis	−0.020	−0.690	−0.752	−0.655		

Note: * *p* < 0.05, ** *p* < 0.01, *** *p* < 0.001; *M* = mean; *SD* = standard deviation.

**Table 3 behavsci-14-00352-t003:** The indirect effect of self-control on CCC via openness and the PNS.

Path	Standardized Coefficient	95% CI	
		Lower	Upper
SC → Openness → CCC	0.023	−0.002	0.055
SC → PNS → CCC	0.020	−0.006	0.057
SC → Openness → PNS → CCC	0.008	0.000	0.023
Total effect	0.226	0.089	0.363
Direct effect	0.176	0.041	0.311
Total indirect effect	0.050	0.009	0.100

## Data Availability

The raw data supporting the conclusions of this article will be made available by the authors on request.

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
