# Peer review of "Self-Control, Openness, Personal Need for Structure and Compensatory Control Change: A Serial Mediation Investigation"

_behavsci, 2024, doi:10.3390/bs14050352_

Round 1

Reviewer 1 Report

Comments and Suggestions for Authors

The authors examined the COVID-19 pandemic’s impact on control perception, revealing an increase in compensatory control behaviors. They found self-control to be a direct predictor of these behaviors, with openness and personal need for structure acting as mediators. The study emphasizes the importance of self-control in maintaining control during the pandemic and suggests strategies for future crises.

I commend the manuscript for enhancing our understanding of the interplay between self-control, openness, personal need for structure, and compensatory control behaviors during the pandemic. It offers insights into the psychological impacts of the pandemic and potential coping strategies. The manuscript significantly contributes to our knowledge of these complex relationships and their implications. Despite a few shortcomings, which I will address, the topic should be of interest to the readers of Behavioral Sciences.

Issues regarding the theoretical background and discussion sections

Despite the theoretical foundation of the research questions, I missed some key findings on the links between trait self-control and individuals’ experiences/behavior during the COVID-19 pandemic, which need to be addressed for the interpretation of the present findings. First, the authors assumed that trait self-control is negatively related to compensatory control change, which might explain individuals’ higher consistency in life structure and behavior during the COVID-19 pandemic (i.e., behavior remains as it was before the pandemic). In my view, this hypothesis is not necessarily plausible, since studies found that high trait self-control was a key protective factor in the response to the COVID-19 pandemic (Lindner et al., 2022). In their study, Lindner et al. (2022) found that high trait self-control was related to lower levels of perceived risk of infection which in turn was associated with lower health anxiety. Furthermore, individuals with high levels of self-control showed higher levels of stress tolerance, protecting them from dysfunctional COVID-19 health anxiety as well. These findings might explain the positive direct effect of self-control on compensatory control change in the present study (mediation analysis). One argument is that high self-controlled individuals might have focused more strongly on the regulation of their behavior in line with governmental restrictions (e.g., social distancing, wearing masks and staying at home). When their priority was to follow the restrictions (and therefore, to avoid increases in health anxiety), they might have also used some compensatory control strategies such as stocking food or toiletries. Furthermore, high self-controlled individuals might have also increased their information-seeking behavior (in social media) to get updates on how to behave, which in turn might have led to increasing probabilities for (spreading) misinformation. Taken together, these findings of the present study need to be interactively discussed against the background of the findings (e.g.) of Lindner et al. (2022).

Issues regarding the discussion of the result

The authors provide inconsistent findings in the results section that need to be discussed in more detail. In Table two, they found a negative correlation between self-control and compensatory control change, whereas a positive relation between both variables was found in the mediation analysis (see Figure 1). The inconsistency might be due to suppression effects (or something else). I think the paper might benefit from exploring/discussing potential confounders (without reanalysis of the data).

Once the issues I mentioned above have been resolved, I recommend the manuscript for publication.

References:

Lindner, C., Kotta, I., Marschalko, E. E., Szabo, K., Kalcza-Janosi, K., & Retelsdorf, J. (2022). Increased Risk Perception, Distress Intolerance and Health Anxiety in Stricter Lockdowns: Self-Control as a Key Protective Factor in Early Response to the COVID-19 Pandemic. International Journal of Environmental Research and Public Health, 19(9), 5098. https://doi.org/10.3390/ijerph19095098

Reviewer 2 Report

Comments and Suggestions for Authors

The article investigates the relationship between self-control and changes in compensatory control, mediated by openness and personal need for structure, in the context of the COVID-19 pandemic. Investigating these complex relationships seems important to me. However, I do have several issues with the article that the authors need to address:

Open science: Did the authors preregister anything in relation to this study? I was not able to find any mention of this in the article. If nothing was preregistered, why not? The authors would need to address this.

Study design/procedure/participants: When reading the article, I wondered whether this analysis was part of a larger study, perhaps investigating other variables. The study design/procedure/participants were only very briefly mentioned, and I would urge the authors to provide a bit more detail here. For example, why is the sample size substantially higher than what the power analysis suggested?

Measurement of CCC: How were these items selected? For example, the authors mention in the introduction different forms of compensatory behaviors in relation to the COVID-19 pandemic: “[…]such as embracing conspiracy beliefs, preference for order-providing scientific theories, moral arguments[…]” (p. 2, line 47). Why were they not included in this measure? Can the authors provide any data that supports the reliability and validity of this measure?

One time point? It was not clear to me if the data assessment was done in one time point and whether the compensatory behavior “before the outbreak” was therefore retroactively assessed. If so, this would be a limitation that needs to be discussed.

Correlational data: The authors performed a mediation analysis with (if the aforementioned assumption is correct) cross-sectional data. There are several problems associated with this that need to be discussed (e.g. Kline, 2015).

Other: The additional material was not in English, so I cannot comment on it.

References:

Kline, R. B. (2015). The mediation myth. Basic and Applied Social Psychology, 37(4), 202-213. https://doi.org/10.1080/01973533.2015.1049349
